# Insight into the Propagation of Interface Acoustic Waves in Rotated YX-LiNbO_3_/SU-8/Si Structures

**DOI:** 10.3390/mi16080861

**Published:** 2025-07-26

**Authors:** Cinzia Caliendo, Massimiliano Benetti, Domenico Cannatà, Farouk Laidoudi

**Affiliations:** 1Institute for Photonics and Nanotechnologies, IFN-CNR, Via del Fosso del Cavaliere 100, 00133 Rome, Italy; 2Institute of Microelectronics and Microsystems, IMM-CNR, Via del Fosso del Cavaliere 100, 00133 Rome, Italy; massimiliano.benetti@cnr.it (M.B.); domenico.cannata@cnr.it (D.C.); 3Research Center in Industrial Technologies CRTI, Cheraga, P.O. Box 64, Algiers 16014, Algeria; f.laidoudi19@gmail.com

**Keywords:** SAW, LSAW, IAW, LiNbO_3_, SU-8, Si, Pt, IDTs

## Abstract

The propagation of interface acoustic waves (IAWs) along rotated YX-LiNbO_3_/SU-8/ZX-Si structures is theoretically investigated to identify the Y-rotation angles that support the efficient propagation of low-loss modes guided along the structure’s interface. A three-dimensional finite element analysis was performed to simulate IAW propagation in the layered structure and to optimize design parameters, specifically the thicknesses of the platinum (Pt) interdigital transducers (IDTs) and the SU-8 adhesive layer. The simulations revealed the existence of two types of IAWs travelling at different velocities under specific Y-rotated cuts of the LiNbO_3_ half-space. These IAWs are faster than the surface acoustic wave (SAW) and slower than the leaky SAW (LSAW) propagating on the surface of the bare LiNbO_3_ half-space. The mechanical displacement fields of both IAWs exhibit a rapid decay to zero within a few wavelengths from the LiNbO_3_ surface. The piezoelectric coupling coefficients of the IAWs were found to be as high as approximately 7% and 31%, depending on the Y-rotation angle. The theoretical results were experimentally validated by measuring the velocities of the SAW and LSAW on a bare 90° YX-LiNbO_3_ substrate, and the velocities of the IAWs in a 90° YX-LiNbO_3_/SU-8/Si structure featuring 330 nm thick Pt IDTs, a 200 µm wavelength, and a 15 µm thick SU-8 layer. The experimental data showed good agreement with the theoretical predictions. These combined theoretical and experimental findings establish design principles for exciting two interface waves with elliptical and quasi-shear polarization, offering enhanced flexibility for fluidic manipulation and the integration of sensing functionalities.

## 1. Introduction

Lithium niobate (LiNbO_3_) is a well-known piezoelectric material particularly significant in the fabrication of electroacoustic devices, such as filters, resonators, delay lines, sensors, and actuators, for applications in telecommunications, photonics, microfluidics, and sensing fields [1] due to its remarkable electro-optical and piezoelectric properties. The propagation of surface acoustic waves (SAWs) has been extensively explored for different LiNbO_3_ crystallographic cuts and wave propagation directions [2,3]. Some rotated YX-LiNbO_3_ cuts can sustain the propagation of both SAWs and leaky SAWs: The velocity of the latter waves can be larger than that of the former (quite near the velocity of the Shear Horizontal Bulk Acoustic Wave, SHBAW), thus allowing higher operating frequencies. Moreover, the electromechanical coupling coefficient *K*^2^ of the LSAWs can be larger than that of the SAWs, as described in reference [4].

The remarkable *K*^2^ values observed for SAWs and LSAWs in rotated Y-cuts of LiNbO_3_ prompt further investigation into their potential application in electroacoustic devices based on the propagation of interface acoustic waves (IAWs). Also known as Stoneley waves, IAWs propagate along the planar boundary between two perfectly bonded solids. Their velocity lies between that of the SAW in the substrate and the lowest bulk acoustic wave (BAW) velocity in the overcoat [5,6], and their displacement amplitude decays to zero on both sides of the interface. However, only a limited number of substrate–overcoat combinations support the existence of IAWs; a more detailed discussion of this topic is provided in references [7,8,9].

Interest in IAWs first emerged in the early 1980s in connection with the development of packageless devices [7,10,11], owing to the advantages they offer over conventional SAW devices—such as radio-frequency filters and resonators—which typically require encapsulation within a hermetic package to shield them from environmental disturbances (e.g., dust, humidity, radiation, or even minor fluctuations in ambient pressure and temperature) and to ensure sufficient clearance between the device surface and the enclosure. In contrast, IAW-based devices do not require cavity formation, as the wave energy is confined to the interface between two bulk materials. Moreover, their manufacturing process is comparable to that of standard SAW filters or resonators, and IAWs can be excited and detected using interdigital transducers (IDTs).

In the early 2000s, IAW-based microfluidic devices for particle manipulation were introduced as a means to overcome the limitations of SAW-based systems, as demonstrated in reference [12]. Conventional SAW-driven microfluidic platforms [13,14,15] typically utilize Rayleigh-type surface acoustic waves on 128° Y-cut LiNbO_3_ substrates, known for their high electromechanical coupling coefficient (*K*^2^) and substantial mechanical displacement amplitudes. These systems often incorporate polydimethylsiloxane (PDMS) microchannels with both side and top walls. However, as SAWs propagate beneath the PDMS structures, they experience considerable attenuation due to acoustic energy absorption by the polymer. This energy loss significantly reduces the amount of acoustic power that can be effectively delivered into the fluidic region, thereby limiting device performance.

The unique characteristics of the IAW—their confinement at the interface and their sensitivity to mechanical properties—make them promising candidates for enhancing the performance of advanced electroacoustic devices. These devices can be employed in both well-established and emerging fields, such as (i) study of the mechanical properties of interfaces (e.g., detection of defects or delamination in multilayer structures); (ii) microfluidics and biosensing (devices for manipulating nanoparticles or molecules in nanofluidic systems, assuming the presence of localized channels excavated into the overcoat); (iii) MEMS applications in advanced acoustic devices, with a focus on miniaturization (package-less devices) and enhanced stability (embedded devices); (iv) sensors of liquid media conductivity or viscosity, sensors of gas concentration (under the hypothesis that a gas flows inside a microchannel excavated in the overcoat), and optical sensors (under the hypothesis that the overcoat and/or the substrate is transparent to the optical radiation to be revealed).

In the present paper, the characteristics of IAWs propagating in multilayered structures composed of a rotated YX-LiNbO_3_ substrate and a Si overcoat material, bonded together by a SU-8 adhesive layer, were theoretically investigated for different SU-8 layers and metal electrode thicknesses with respect to the phase velocity, propagation losses, and electromechanical coupling efficiency. The present study demonstrates that the rotated YX-LiNbO_3_ bare half-space supports the propagation of both SAW and LSAW modes, exhibiting relatively high electroacoustic coupling efficiencies up to approximately 6% and 23%, respectively. Furthermore, when the piezoelectric half-space is coupled with an SU-8/Si overcoat, it supports the propagation of two IAW modes, which also show strong electroacoustic coupling efficiencies, up to around 7% and 24%.

In accordance with theoretical predictions, the propagation of the SAW and LSAW was experimentally assessed in the bare 90°-rotated YX-LiNbO_3_ half-space, as well as the propagation of two IAWs in the 90°-rotated YX-LiNbO_3_/SU-8/Si layered structure. The measured wave velocities were found to be in good agreement with the corresponding theoretical values.

## 2. Methods

The three-dimensional finite element method (3D FEM) was employed to perform eigenfrequency and frequency domain analysis using Comsol Multiphysics 6.2 software. The models use the piezoelectric multiphysics coupling node comprising solid mechanics and electrostatic interfaces. The rotated YX crystal cut was obtained by the application of a rotated coordinate system based on the Euler angles and defined as the rotations from the material crystallographic axis to the reference axis, following the Z−X−Z convention. For a rotated YX-LiNbO_3_, the Euler angles (Ф Ψ Θ) are Ф = Θ = 0 and Ψ (Ψ represents the counterclockwise rotation angle of the Z-axis around the X-axis), as shown in Figure 1. In simulations, a wavelength λ = 200 µm was assumed.

The propagation of SAWs in rotated YX-LiNbO_3_ half-spaces and IAWs in rotated YX-LiNbO_3_/SU-8/Si structures has been studied in terms of the wave velocity, propagation loss, and electromechanical coupling coefficient.

Figure 2a shows the primitive 3D SAW cell, which consists of three domains: (i) a LiNbO_3_ domain (height 5λ, width λ = 200 µm); (ii) two Pt IDTs with a width equal to λ/4 and a thickness of 0.15 µm (IDTs were not considered for eigenfrequency domain analysis); and (iii) an air domain (height λ, width λ = 200 µm). The depth is assumed to be λ/5 for all the domains. The two metal IDTs are positioned on the top surface of the LiNbO_3_ substrate for frequency domain analysis: one electrode is grounded, the other is connected to a fixed electric potential of 1 V. The air, Pt, and LiNbO_3_ material constants are those reported in the Comsol library. Since the metal IDTs are much thinner than the other domains, numerical modelling of such a thin layer in 3D requires very small meshes, leading to high computational cost and time. To avoid this issue, the thin layer option of Comsol was used to represent the metal IDTs, thus allowing a substantial reduction in computational cost and time, without sacrificing the accuracy of the simulation. Beneath the LiNbO_3_ domain, a perfectly matched layer (PML) with one wavelength height was added to capture losses related to bulk wave radiation [16,17]. Periodic boundary conditions were applied at the lateral sides of the unit cell, parallel to the X and Y directions, to emulate an infinitely long resonator with an infinite number of electrodes; free and fixed conditions were applied, respectively, to the top and bottom surfaces of the structure. An extremely fine mesh (physics-controlled mesh, maximum and minimum element size of 52 and 0.52 µm) was chosen for all the FEM simulations.

Since damping loss factors for SU-8, LiNbO_3_, Pt, and silicon are not readily available in the literature, a generic damping (isotropic loss factor) of 10^−4^ was attributed to the LiNbO_3_ half-space (with also permittivity loss factor of 10^−4^) and the Pt layer to prevent infinite gain at resonance in the undamped system.

Figure 2b shows the 3D unit cell adopted to study the propagation of the IAWs: it consists of the PML, LiNbO_3_ substrate (1000 µm thick), the SU-8 layer (with variable thickness), and the overcoat (500 µm thick). The two metal interdigitated electrodes (λ/4 wide) have variable thickness (from 0.15 up to 1.0 µm); the same damping (isotropic loss factor) of 10^−4^ was attributed to the LiNbO_3_ half-space (also with a permittivity loss factor of 10^−4^) and the SU-8 and Si overcoat materials.

## 3. Eigenfrequency Domain Analysis

Eigenfrequency domain analysis was performed to calculate the natural frequencies of the modes propagating along the surface of the bare piezoelectric substrate and the interface of the LiNbO_3_/SU-8/Si structure. The rotated YX-LiNbO_3_ half-space can sustain the propagation of both SAWs and LSAWs depending on the rotation angle, as described in reference [3]: the former wave is characterized by elliptical polarization, and its velocity is smaller than that of the LSAW, which travels at a velocity between that of the fast and slow shear BAWs. Each BAW can transform into the quasi-bulk surface wave solution (LSAW) or into the unphysical solution, depending on the cut angle. A similar behaviour is obtained for IAWs.

Figure 3a shows the angular dispersion curves of the phase velocity for the SAW and LSAW propagating on the bare piezoelectric substrate (black and red solid lines), and the lower and upper IAWs (black and red dashed lines) propagating along the interface of the LiNbO_3_/SU-8/Si structure. As highlighted in Figure 3a, when approaching the −38° rotation angle (corresponding to the 128° YX cut), the SAW solutions become unphysical and are therefore not reported in Figure 4a. As shown in Figure 4a, the phase velocity of the SAW increases as it evolves into the lower IAW, whereas the phase velocity of the LSAW decreases as it transitions into the upper IAW.

In Figure 3b, the angular dispersion curves of the electromechanical coupling coefficient (*K*^2^) are reported for the different modes. *K*^2^ was calculated by using the well-known formula:(1)$$K^{2}=2\frac{∆v}{v}=2\frac{v_{OC}-v_{SC}}{V_{OC}}$$
where $v_{OC}$ refers to an open-circuit electrical condition and $v_{SC}$ refers to a short-circuit electrical condition applied on the top surface of the LiNbO_3_ substrate. The SAW exhibits a 5.8% maximum *K*^2^ value (approaching 128° YX cut, i.e., Ψ = −38°), while the LSAW can reach a *K*^2^ as high as about 23% (for Ψ = 76°). The present results are consistent with those reported in the literature [3,4], with the exception of a slight discrepancy in the maximum *K*^2^ values—5.4% and 21% for the SAW and LSAW, respectively. This deviation can primarily be attributed to differences in the LiNbO_3_ material constants adopted in the calculations. Notably, significant *K*^2^ values are also obtained for the lower and upper IAWs, only slightly lower than those of the SAW and LSAW modes.

The wave propagation losses were evaluated as(2)L=2π·8.686·(fimagfreal)
where $f_{imag}$ and $f_{real}$ are the imaginary and the real part of the mode eigenfrequency. The angular dispersion curves of the propagation losses are shown in Figure 3c. As expected, the propagation losses for the SAW and the lower IAW are significantly lower than those of the LSAW and upper IAW, due to the absence of energy leakage into the bulk. Nevertheless, the propagation losses for the LSAW and upper IAW remain very low over a wide range of rotation angles (i.e., 10° < Ψ < 90°), while maintaining high *K*^2^ values. These characteristics suggest their potential suitability for applications in electroacoustic devices for telecommunications and sensing.

Figure 4 shows the solid displacement of the waves propagating on 76°-rotated LiNbO_3_ for the case of a bare piezoelectric substrate and the structure LiNbO_3_/SU-8/Si, as an example. Figure 4a,b show the solid displacement of the SAW and LSAW; Figure 4c,d show the solid displacement of the lower and upper IAW. The SAW and lower IAW are elliptically polarized, while the LSAW and upper IAW are predominantly shear-horizontally polarized. Figure 4e,f show the shape of the mechanical field—the solid displacement vs. the depth—for each IAW; it can be noticed that the mechanical displacement field of both IAWs exhibits a non-null but rapid decay to zero from the LiNbO_3_ surface toward the Si overcoat. The acoustic field of the lower IAW penetrates the LiNbO_3_ substrate up to about one wavelength in depth, while that of the upper IAW decays in the LiNbO_3_ substrate within a few wavelengths.

The SAW exhibits a maximum *K*^2^ of 6% at the 128° YX cut, while the LSAW reaches a *K*^2^ as high as approximately 32% at the 82° YX cut. The present calculations are in good agreement with the results reported in references [3,4], except for a discrepancy in the maximum *K*^2^ values (5.4% and 21% for SAW and LSAW, respectively), which may be attributed to differences in the LiNbO_3_ material constants, the calculation method adopted, and its numerical accuracy. The remarkable *K*^2^ values and the distinct polarization characteristics of both the SAW and LSAW suggest their potential for application in electroacoustic devices for telecommunications and sensing.

## 4. IAWs in Rotated YX-Propagation LiNbO_3_/IDT/SU-8/Si

Additionally, 3D FEM calculations were performed to study the propagation of IAWs in rotated YX-LiNbO_3_/SU-8/Si structures for different thicknesses of the SU-8 layer and Pt IDTs. The SU-8 is inserted in the studied multilayer structure for its role as an adhesive element between the overcoat on the piezoelectric substrate. Unfortunately, however, it can influence the propagation characteristics more or less heavily depending on its thickness: the effects of its thickness on the acoustic propagation characteristics have been studied over a wide range (from 5 up to 25 µm), but in the end, it was decided to fix its thickness at 15 µm to avoid the propagation of unwanted acoustic modes inside the SU-8 layer, but also obtain uniform and repeatable spin-coated layers.

The thickness of the Pt IDTs was varied within a narrow range (0.15 to 1.0 µm). Although simulations indicated that Pt layers thicker than 1.0 µm significantly enhance the performance of IAW-based devices by improving electromechanical transduction, a lower thickness (<1.0 µm) was selected to minimize parasitic capacitance and RF losses. This choice also ensures compatibility with nanometre-scale lithographic processes, thereby enabling cost-effective fabrication.

### Frequency Domain Study

A frequency domain study was conducted to extract the resonant frequencies from the admittance Y_11_ vs. frequency curves of the IAW-based resonators. The curves were computed for Y-rotation angles (Ψ) of LiNbO_3_ ranging from −90° to 90° in 2° steps, for different Pt IDT thicknesses (0.15, 0.5, 0.7, and 1.0 μm), and over a frequency range from 18 to 25 MHz with a 0.001 MHz step. The results demonstrate that the admittance versus frequency curves for both modes are influenced not only by the Y-rotation angle but also by the IDT thickness. This behaviour is consistent with that observed for SAWs and leaky SAWs propagating in rotated YX-LiNbO_3_ half-spaces, as reported in references [3,4]. As an illustrative example, Figure 5 shows the resonant frequency versus rotation angle curves for the two IAW modes with IDT thicknesses of 0.15 μm and 1.0 μm. The colour bar indicates the absolute value of the admittance, |Y_11_|. These results further confirm the correlation between the Y_11_ frequency response and the *K*^2^. As the IDT thickness increases, the mode velocity—and consequently the resonant frequency—decreases due to the mass loading effect.

From Figure 5, it can be noticed that the LiNbO_3_/SU-8/Si structure can sustain the propagation of two IAWs at specific Y-rotation angles: one IAW travels at a velocity a little larger than that of the SAW in bare LiNbO_3_ half-space, while the other one travels at a velocity a little lower than that of the LSAW in bare LiNbO_3_ half-space. In a certain range of rotation angles, both the LSAW and upper IAW cross the BAW velocity and, hence, become unphysical solutions, in accordance with the numerical results shown in reference [3,4]. These results are in accordance with the eigenfrequency domain analysis previously reported in Section 3 of this work.

The phase velocity of the IAWs was calculated as $v_{ph}$ = $\lambda \cdot f_{res}$ being $f_{res}$ the abscissa of the real (Y_11_) admittance peak. The effective coupling coefficient *K*^2^, which represents the electrical-to-acoustic energy conversion efficiency by electrical means, was estimated by using the following formula:(3)$$K^{2}=\frac{\pi^{2}}{4}\cdot \frac{f_{a}-f_{r}}{f_{a}}$$
where $f_{a}$ and $f_{r}$ are the antiresonance and resonance frequencies extrapolated from the abs (Y_11_) vs. frequency curves at each Y-rotation angle. The phase velocity and *K*^2^ of the lower IAW were calculated at different Y-rotation angles, for different Pt IDT thicknesses (0.15, 0.5, and 1.0 µm). Figure 6a,b show the angular dispersion curve of the phase velocity and *K*^2^ of the lower IAW: with increasing the Pt thickness, the IAW velocity decreases due to the mass loading effect; the *K*^2^ of the peak at Ψ = −42° increases while that at Ψ = 90° weakly decreases.

Figure 7a,b show the angular dispersion curves of the phase velocity and *K*^2^ of the upper IAW, for different Pt IDT thicknesses (0.15, 0.5, and 1.0 µm). For the eigenfrequency study, approaching the −38° rotation angle (128 °YX cut), the solutions became unphysical, and they are not reported in Figure 7. With increasing the Pt IDTs thickness, the phase velocity decreases due to the mass loading effect, and the *K*^2^ improves since the wave energy entrapment is enhanced by the thicker Pt IDTs in accordance with [3,4].

The lower IAW shows a good but significantly smaller effective coupling coefficient than the fast IAW, which instead shows a *K*^2^ that can be significantly increased simply by properly designing the electrode thickness and optimizing it for an appropriate choice of the rotated Y-axis. A thicker Pt IDT guarantees better performance: the performed calculations suggest that a quite remarkable *K*^2^ can be obtained for IAW propagating in YX-LiNbO_3_/SU-8/Si (about 29%), even if 32% can be reached at Ψ ≈ 78° rot. Y, for Pt IDT 1 µm thick.

The impact of different IDT materials on the performance of the IAW-based resonator in the YX-LiNbO_3_/SU-8/Si structure was analyzed. The S_11_ scattering parameter was calculated for Al, Pt, Ti, Cr, and Au, which are used as electrode materials. The IDTs had a fixed thickness of 0.15 µm, which is compatible with conventional SAW device fabrication processes. Figure 8 shows the S_11_ vs. frequency curves for different metal electrodes calculated as $S_{11}=20\cdot log\left(\frac{-abs\left(Y\right)+1/50}{abs\left(Y\right)+1/50}\right)$: two IAWs are visible at frequencies around 19 and 20.4 MHz.

The S_11_ of the fast mode exhibits a peak with an amplitude more than twice that of the slow mode, in agreement with the angular dispersion curves of *K*^2^. According to Figure 8, Pt appears to be the most suitable material for fabricating the IDTs intended to predominantly excite the upper IAW, while the lower IAW seems to be largely unaffected by the choice of electrode material.

## 5. Experimental Section

### 5.1. Devices Fabrication

A SAW delay line was patterned onto the surface of a doubly polished YX-LiNbO_3_ substrate (“4” diameter and 1 mm thick) by a lift-off process [18,19]; a Pt film, 0.33 µm thick, was grown by the sputtering technique at 200 W for 5 min (at ambient temperature). Although the calculations performed demonstrated that Pt films thicker than 1 µm enhance the piezoelectric coupling, sub-micrometre-thick IDTs were selected in order to reduce fabrication costs. The delay line has a 200 µm wavelength in a single-finger configuration (number of finger pairs *N* = 18, IDTs centre to centre distance L = 7750 µm, fingers overlapping 8500 µm). The device was fixed onto a printed circuit board (PCB) with an epoxy adhesive; the pads of the IDTs were electrically connected to the SMA connectors fixed onto the PCB by Al wires soldered by ultrasonic bonding. Figure 9a shows a photo of the delay line patterned onto the LiNbO_3_ substrate and ready to be tested, i.e., the contact pads have been wired. A 15 µm thick layer of SU-8 (3025 negative epoxy photoresist from MicroChem Corporation) was spin-coated onto a Si substrate at 6000 rpm, following the procedure outlined in [20]. Then the SU-8/Si overcoat was put atop the entire delay line (including the two IDTs and the acoustic wave path) and held under pressure for 10 min at 95 °C. Figure 9b shows the IAW-based devices ready to be tested, i.e., the SU-8 bonding has been achieved, and the IDTs’ pads have been ultrasonically wire-bonded to corresponding pads on the PCBs interfaced with the testing equipment via SMA-type connectors. The adhesion between the overcoat and the piezoelectric substrate was found to be excellent, as expected given the considerable SU-8 thickness (15 μm) and the absence of surface contamination; the fabrication process was carried out in a cleanroom environment in accordance with standard protocols. This was confirmed by the failure of tensile tests aimed at delaminating the overcoat from the substrate. Furthermore, as detailed in the following paragraph, the degree of contact at the interface and the integrity of the bond were assessed by measuring the velocity of the propagating waves: velocities approaching that of the interfacial acoustic wave (IAW)—which are higher than those of surface acoustic waves (SAWs)—indicate successful excitation and detection of IAWs, and thus confirm strong adhesion between the overcoat and the substrate.

### 5.2. Devices Test

The electrical characterization of the delay-line devices was performed by measuring the scattering parameters S_11_ and S_21_ in the frequency domain by means of a vector network analyzer (DG8SAQ VNWA 3 1.3 GHz Vector Network Analyzer, SDR KITS LIMITED UK Company, Melksham, UK) connected to the PC for real-time acquisition. The measured scattering parameters refer to untuned devices without adding any impedance matching. The VNA was calibrated (full two ports calibration) up to the coaxial cables; thus, the parasitic components—associated with the SMA connectors, wire bonds and PCB tracks—were excluded from the calibration.

Figure 10a,b show the S_21_ vs. the frequency curve of the SAW and LSAW travelling in the bare LiNbO_3_ substrate at 18.48 and 22.21 MHz. The measurements were performed both with and without an adsorber (transparent nail polish) applied with a brush along the edges of the piezoelectric substrate (not shown in the photos). It was observed that the absorber effectively suppressed unwanted reflections in the SAW S_21_ signal, resulting in a cleaner and more ripple-free transmission response. Conversely, the adsorber had no noticeable effect on the S_21_ of the LSAW, thus confirming the bulk-like nature of this wave.

Figure 11a,b show the S_21_ vs. frequency curve of the two IAWs (excited at 18.78 and 21.4 MHz) travelling in the LiNbO_3_/SU-8/Si substrate; Figure 11c shows the S_11_ return loss spectrum of both the devices (the bare LiNbO_3_ substrate and the LiNbO_3_/SU-8/Si substrate).

By comparing the S_21_ curves in Figure 10 and Figure 11, it can be observed that the SAW exhibits lower insertion losses than the LSAW, and a similar trend is seen for the two IAWs, consistent with their respective electromechanical coupling coefficients (*K*^2^). Figure 11c shows that the SAW and LSAW peaks in the S_11_ curve are clearly distinguishable and well separated; the SAW peak is significantly smaller than that of the LSAW, and the same applies to the S_11_ peaks of the lower and upper IAWs, reflecting the differences in their *K*^2^ values. The SAW resonance peak is also more pronounced than that of the lower IAW, which displays higher out-of-resonance losses, suggesting a more leaky propagation medium. Furthermore, the S_11_ peak of the upper IAW is more pronounced than that of the LSAW, indicating better coupling between the IDT and the medium, and lower associated losses.

All the scattering parameters were obtained by using time gating to cancel the spurious time signals due to reflections between the IDTs and at the substrate edges, and acoustic energy coupling into BAWs, which are reflected off the backside of the substrate and interfere with the device surface.

In accordance with theoretical predictions, the propagation of the SAW and LSAW was experimentally evaluated in the bare 90°-rotated YX-LiNbO_3_ half-space, as well as the propagation of two IAWs in the 90°-rotated YX-LiNbO_3_/SU-8/Si layered structure. The measured wave velocities exhibited good agreement with the corresponding theoretical values.

## 6. Discussion

The literature on IAW-based electroacoustic devices dates back to 1978, when the first experimental results on IAW propagation were obtained using a LiTaO_3_ substrate covered with a thick SiO_2_ layer, which also served as a passivation layer [21]. In 1983, Shimizu and Irino [6] were the first to propose the fabrication of package-less devices based on the propagation of Stoneley waves at the interface between two bulk substrates. Since then, IAW propagation has been investigated in a variety of structures employing different types of piezoelectric substrates, including quartz [22], 10°- and 0°-rotated YX-LiNbO_3_, 36° YX-LiTaO_3_ [23], 0° and 128° Y-cut LiNbO_3_ [24,25,26], and XZ-LiTaO_3_ [12], to cite just a few.

In reference [12], interfacial acoustic wave (IAW)-based microfluidic devices were proposed as a valid alternative to the SAW-based counterparts. The SAWs typically experience quite large losses—ranging from 6 to 20 dB—when propagating through the walls of the polydimethylsiloxane (PDMS) channel; the PDMS absorbs a significant portion of the acoustic energy, thus limiting the effective acoustic power coupled into the fluidic region. In agreement with the results from the above-mentioned reference, the presently tested devices demonstrated that the acoustic field of the IAWs travelling in LiNbO_3_/SU-8/Si remains sufficiently strong to achieve performance comparable to that of conventional SAW-based devices employing PDMS channels.

While the available literature demonstrates that the IAW-based electroacoustic devices offer attractive features in many application fields (such as package-free filters, resonators, and microfluidics), the present research is aimed at laying the foundation for the design principles of multifrequency IAW-based devices. The present study theoretically predicts and experimentally investigates the propagation of one or two interfacial acoustic waves (IAWs), depending on the Y-rotation angle of the piezoelectric half-space. Specifically, the 90° YX-LiNbO_3_-based multilayered structure supports the propagation of two IAWs, whereas the bare 90° YX-LiNbO_3_ half-space supports the propagation of a surface acoustic wave (SAW) and a leaky surface acoustic wave (LSAW), which exhibit significantly different *K*^2^ values. The insertion loss of the upper IAW (17 dB) is lower than that of the lower IAW (30 dB), due to its approximately three times higher *K*^2^. Furthermore, a previous paper by the authors [21] investigated both theoretically and experimentally the propagation of a single IAW in 128° YX-LiNbO_3_/SU-8/overcoat structures with different overcoat materials (sapphire, SiO_2_, and Si). In contrast, the bare 128° YX-LiNbO_3_ half-space supports the propagation of a single SAW, which corresponds to the highest *K*^2^ value, as shown in Figure 9.

This paper aims to identify efficient excitation conditions through 3D FEM simulations. The eigenfrequency and frequency domain studies occasionally yielded small discrepancies, particularly in the calculation of *K*^2^. This discrepancy arises because the eigenfrequency analysis identifies the natural resonance modes of the system under open-circuit and short-circuit boundary conditions, without accounting for the presence of IDTs or any external electrical excitation. In contrast, the frequency domain study includes the IDT geometry, applies electrical excitation, and computes the steady-state response of the system at a given frequency.

It is worth noting that the present theoretical and experimental study could be further improved by adopting some methodological refinements and complementary validation approaches:

(i)The present calculations could be refined by adopting more accurate material constants and damping loss factors. In particular, reliable loss factors for SU-8, LiNbO_3_, Pt, and silicon are not readily available in the literature. Additionally, the reported material properties of SU-8 exhibit significant variability, with marked discrepancies among different studies. A systematic investigation of the influence of the IDT thickness-to-wavelength ratio and electrode material type would contribute to enhancing the accuracy of the simulations.(ii)Simulations performed on a complete delay-line structure (accounting for both finite IDTs aperture and optimized number of finger pairs) would aid in optimizing the device design.(iii)The tested delay lines were not optimized, but rather fabricated as prototype devices intended to validate the simulation results and not optimized for wave characteristics. Future work will involve fabricating delay lines with different numbers of finger pairs and propagation paths to experimentally measure the mode propagation loss and estimate the associated damping factors.

Table 1 summarizes several strategies that can significantly enhance the performance of the tested devices—each targeting specific design or fabrication aspects to maximize the electromechanical coupling coefficient and overall device efficiency.

The optimized IAW-based device can find application in nondestructive evaluation (NDE) of bonded interfaces (to assess the integrity and adhesion at the interface of two media), sensors optimized for wave propagation inside a buried sensitive interface (i.e., optical radiation sensors based on the acoustoelectric effect), streaming or particle trapping inside localized channels excavated into the overcoat, and so on. IAW-based electroacoustic devices offer attractive features in package-free applications, featuring high performance and a small size to meet application needs in harsh environments.

The excitation of two modes at different frequencies and with distinct polarization—one elliptically polarized and the other quasi-shear-horizontal—offers significant potential for the development of highly sensitive and robust multifunctional devices (e.g., filters, encoders, and complex sensors) within compact footprints. Fast IAW modes are particularly attractive, as they overcome the high-frequency limitations typically encountered in conventional SAW resonators, thereby enabling advanced signal processing. IAWs with different polarizations are well-suited for the development of both sensors capable of operating in harsh environments and actuators for microfluidic applications.

In the presence of a liquid flowing through a microchannel laser-micromachined [27,28] into the Si/SU-8 overcoat, devices based on the lower IAW can be employed for in-liquid operations, such as fluidic actuation, particle manipulation, and sorting, while devices based on the upper IAW can be used to sense the physical properties of the liquid medium. References [29,30,31] are review articles that provide a comprehensive overview of the theory and applications of microfluidic devices. They explore various designs, configurations, and methodologies employed in SAW-based microfluidic systems, offering useful insights that may assist the reader.

## 7. Conclusions

Interface acoustic waves differ from SAWs in that they travel along the interface between two distinct materials. While their energy is distributed across both media, it is primarily confined to the top surface of one medium. As a result, the IAWs are highly confined modes whose propagation characteristics are immune to undesirable environmental effects occurring at the surface of the overcoating crystal; instead, they are primarily influenced by the properties of the two interfacing materials. On the contrary, the SAWs travel along the surface of a solid and are highly sensitive to surface conditions. Like SAW devices, IAW-based devices can be fabricated using planar technologies, allowing for manufacturing costs comparable to those of integrated circuits.

The objective of the present research is to establish theoretical foundations for predicting the performance of IAWs in Y-rotated X-propagation LiNbO_3_/SU-8/Si(001) structures; apply this knowledge to identify the Y-rotation angles that allow the efficient propagation of acoustic modes along the structure’ interface; and optimize device designs in terms of the thickness of both the metal electrodes and the SU-8 adhesive layer. The performed theoretical studies demonstrated that two types of IAWs (at specific Y-rotated cuts of the LiNbO_3_ half-space) can exist, which are piezoelectrically active and show a fast decay with distance from the interface. One of the IAWs exhibits a higher velocity than the SAW propagating on the surface of the bare piezoelectric substrate, and its piezoelectric coupling coefficient can reach approximately 6%. The other IAW is faster than the leaky SAW propagating on the bare surface, and its piezoelectric coupling coefficient can reach up to 30%, depending on the Y-rotation angle. The impact of different electrode materials on the IAW propagation was studied, and Pt was found to be the best choice. The theoretical results were experimentally validated by measuring the scattering parameters, S_11_ and S_21_, of SAWs travelling in the bare 90° Y-X LiNbO_3_ substrate, as well as of the IAWs propagating in the 90° Y-X LiNbO_3_/SU-8/Si structure with Pt IDTs 330 nm thick, an acoustic wavelength of 200 µm, and an SU-8 adhesive layer 15 µm thick.

The commercial availability and cost-effectiveness of both silicon and LiNbO_3_ substrates make them well-suited for the development of acoustofluidic and sensing systems. The co-integration of interface modes with distinct polarizations on a single chip broadens microfluidic capabilities and supports multifunctional operation. The theoretical and experimental results presented in this work provide a foundation for designing devices capable of exciting multiple interface waves, thereby enabling versatile fluid control and on-chip sensing.

## Figures and Tables

**Figure 1 micromachines-16-00861-f001:**
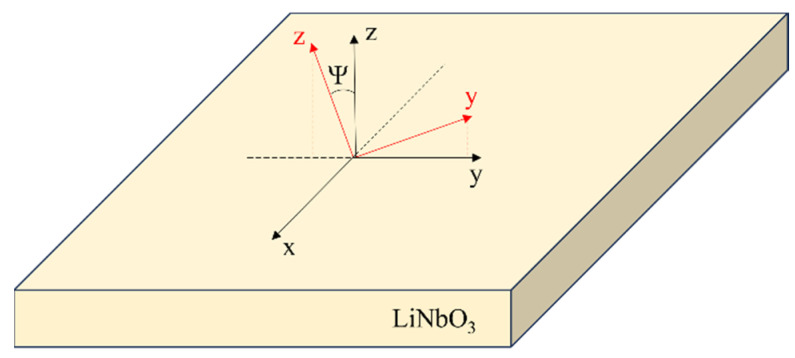
The unrotated (black) and the rotated (red) LiNbO_3_ reference system.

**Figure 2 micromachines-16-00861-f002:**
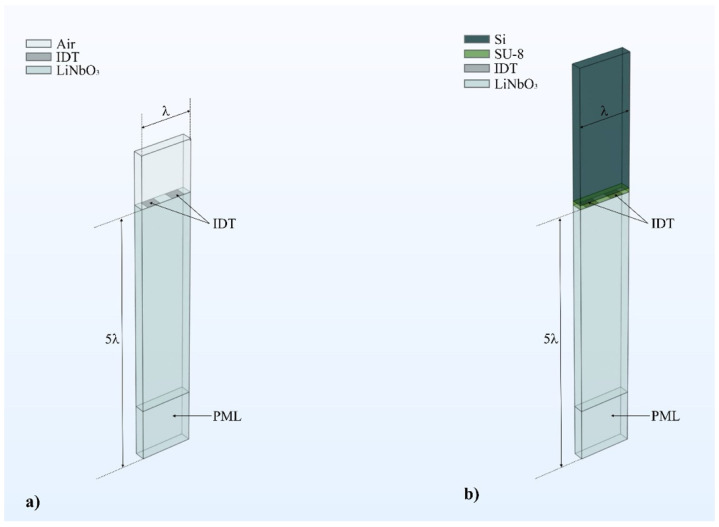
(**a**) The PML/LiNbO_3_/IDT/AIR unit cell employed to study the SAW propagation; (**b**) the PML/LiNbO_3_/IDT/SU-8/Si unit cell employed to study the IAW propagation.

**Figure 3 micromachines-16-00861-f003:**
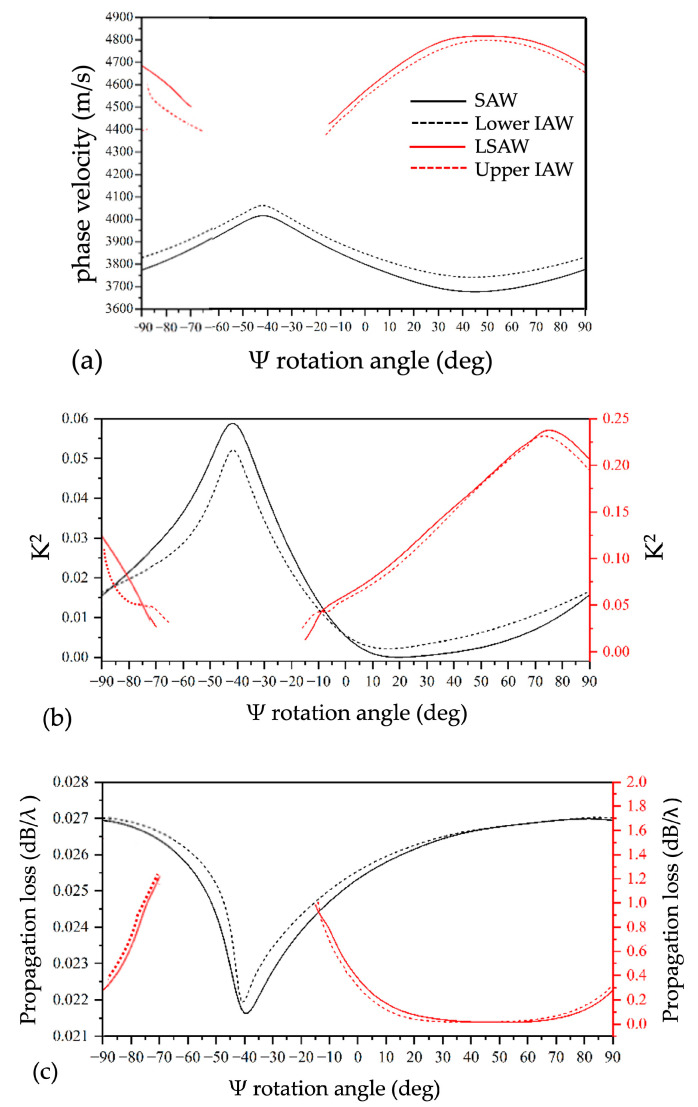
The angular dispersion curves of (**a**) the phase velocity; (**b**) the *K*^2^; and (**c**) the propagation losses for the SAW and LSAW propagating on the bare piezoelectric substrate (black and red continuous lines) and for lower and upper IAWs (black and red dashed lines) propagating at the interface of the structure LiNbO_3_/SU-8/Si.

**Figure 4 micromachines-16-00861-f004:**
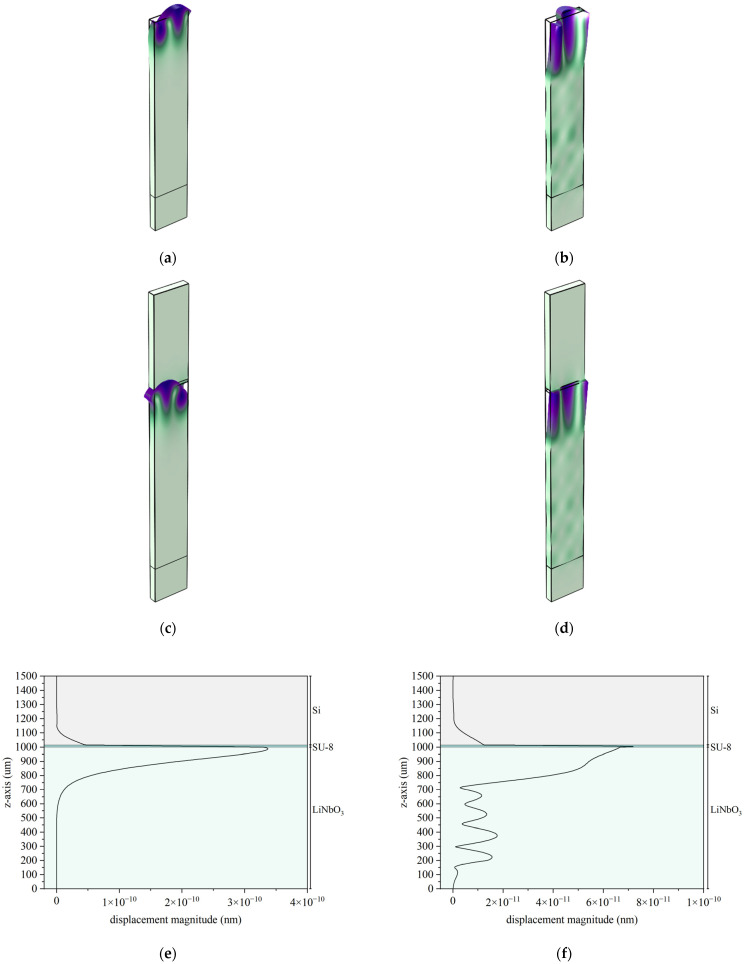
The solid displacement of the (**a**) SAW; (**b**) LSAW; (**c**) lower IAW; and (**d**) upper IAW travelling along 76-rotated LiNbO_3_; the depth profile of the mechanical field of (**e**) the lower and (**f**) upper IAW.

**Figure 5 micromachines-16-00861-f005:**
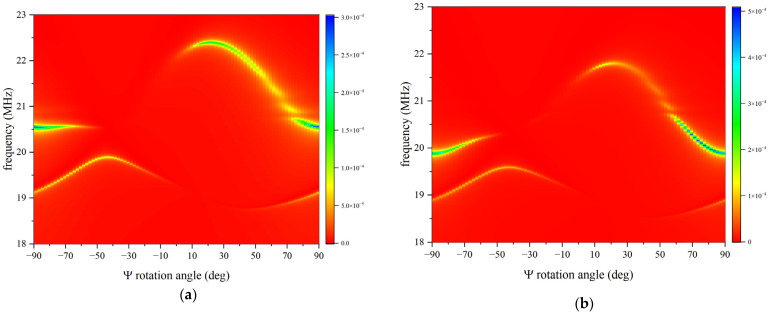
The frequency vs. the Y-rotation angle curves of the two IAWs for Pt IDTs: (**a**) 0.15 and (**b**) 1.0 µm thick; the coloured bar represents the absolute value of Y_11_.

**Figure 6 micromachines-16-00861-f006:**
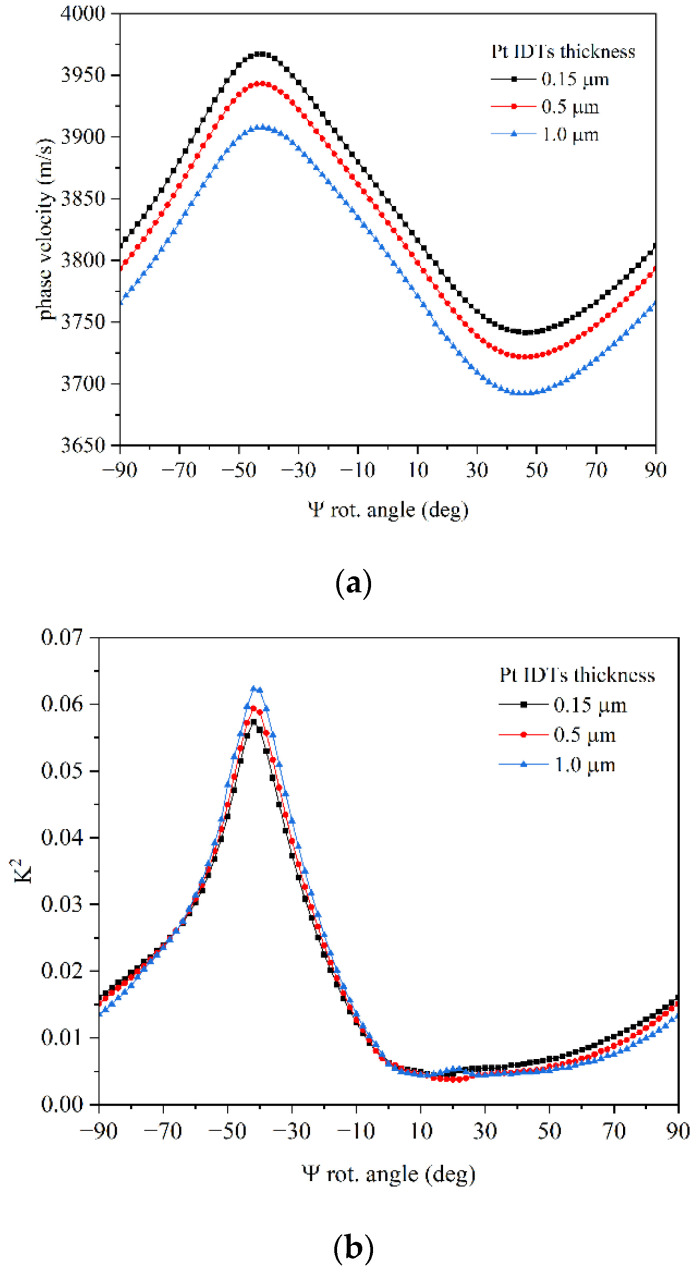
The angular dispersion curves (**a**) of the velocity and (**b**) of the *K*^2^ of the lower IAW for 0.15, 0.5, and 1.0 µm Pt IDT thicknesses.

**Figure 7 micromachines-16-00861-f007:**
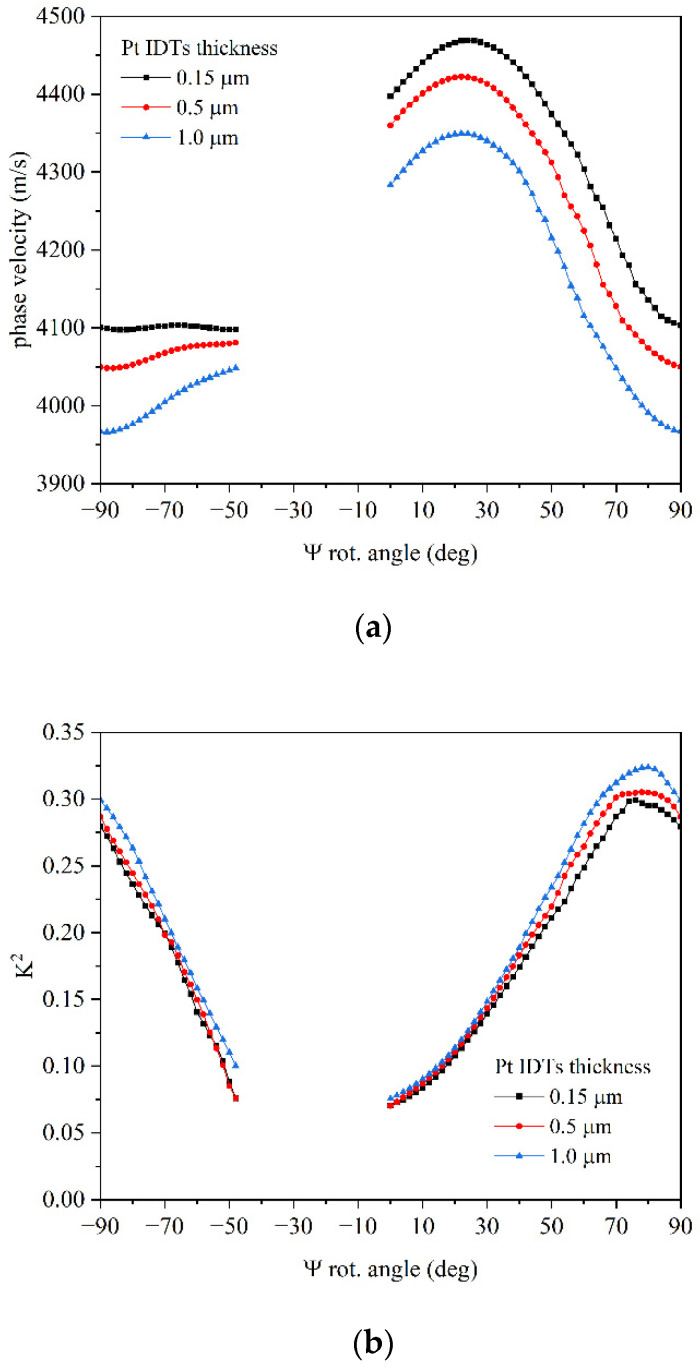
(**a**) The phase velocity and (**b**) the *K*^2^ dispersion curve of the upper IAW for 0.15, 0.5, and 1.0 µm thick Pt IDTs.

**Figure 8 micromachines-16-00861-f008:**
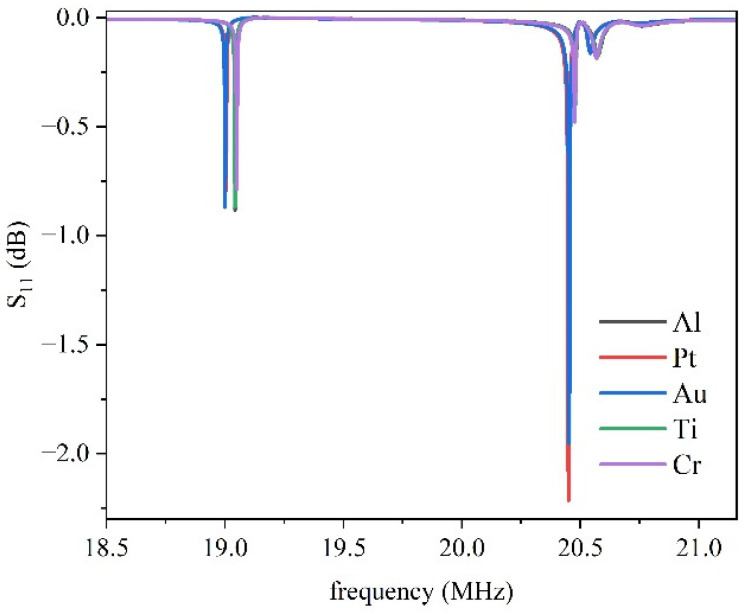
The S_11_ vs. frequency curves for different metal electrodes.

**Figure 9 micromachines-16-00861-f009:**
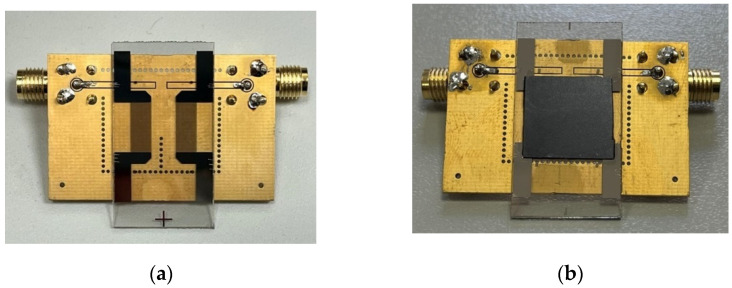
The photo of (**a**) the SAW- and (**b**) the IAW-based delay lines ready to be tested.

**Figure 10 micromachines-16-00861-f010:**
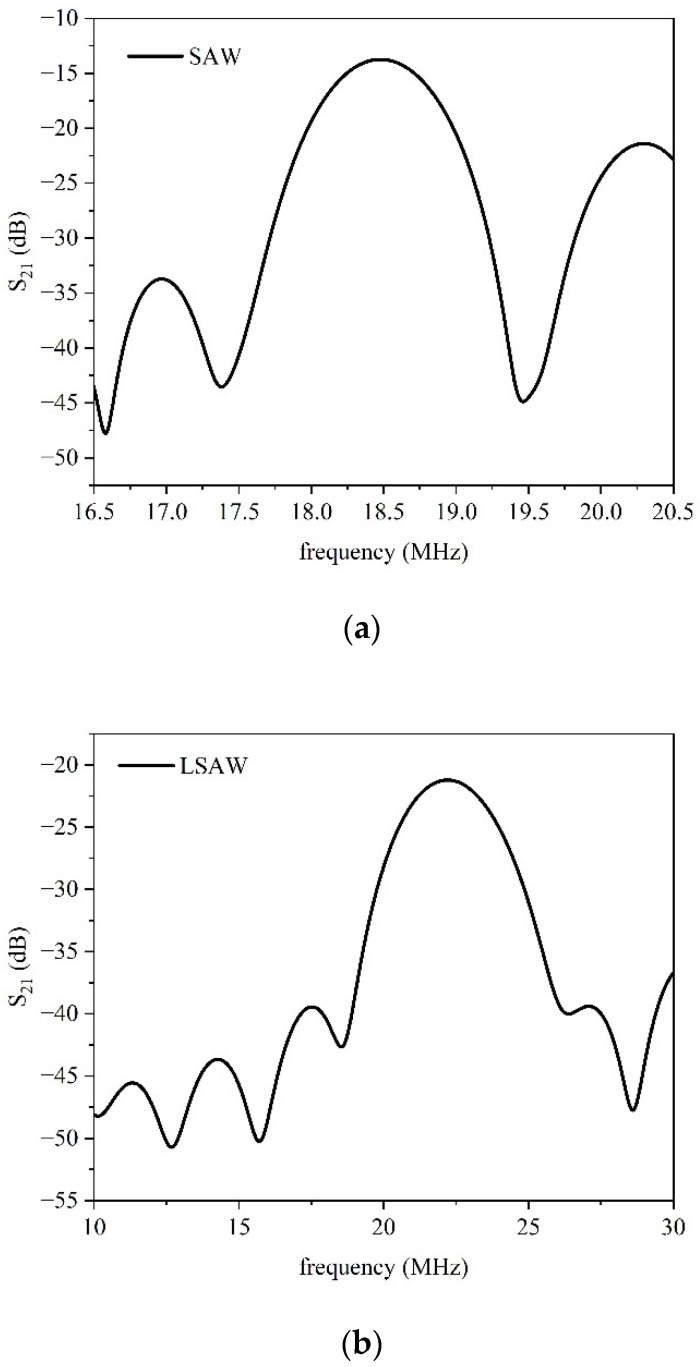
The frequency curve of (**a**) the SAW and (**b**) the LSAW travelling in the bare LiNbO_3_ substrate, referring to the Pt IDTs on the bare LiNbO_3_ substrate.

**Figure 11 micromachines-16-00861-f011:**
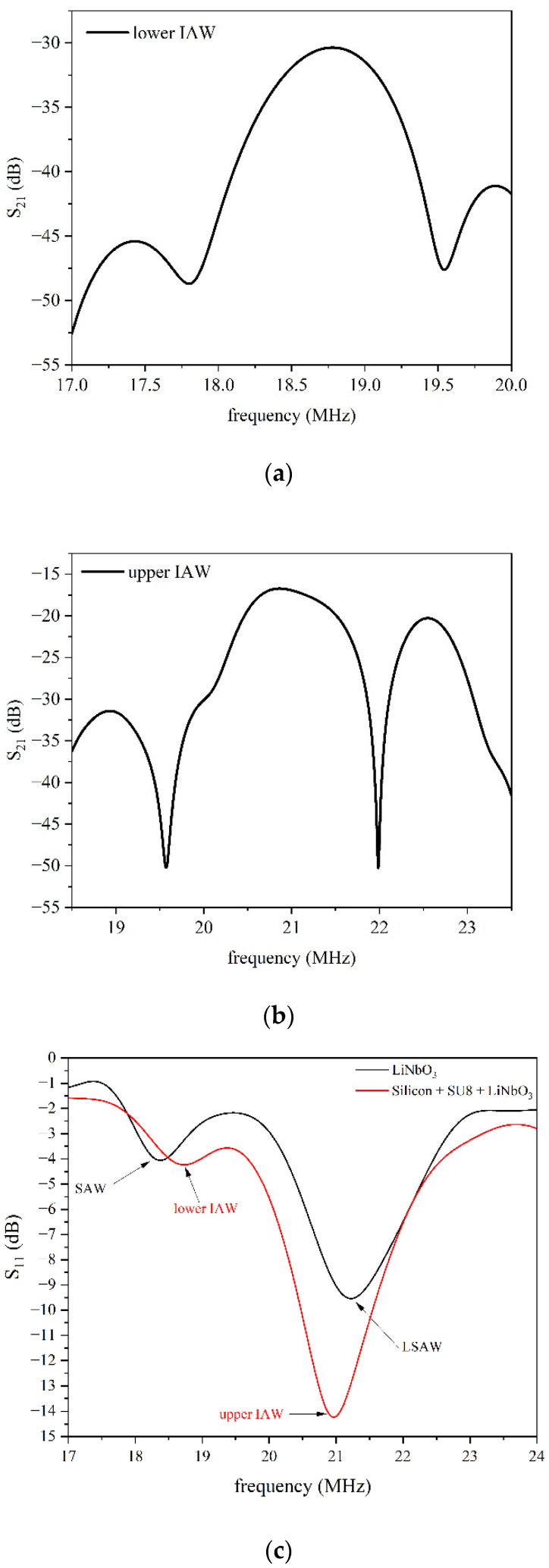
The S_21_ vs. frequency curve of (**a**) the lower and (**b**) upper IAW travelling in the LiNbO_3_/SU-8/Si substrate; (**c**) the S_11_ vs. frequency curves of the bare LiNbO_3_ substrate (black curve) and of the IAWs (red curve) travelling in the LiNbO_3_/SU-8/Si substrate.

**Table 1 micromachines-16-00861-t001:** Summary of optimization strategies for enhancing the performance of LiNbO_3_-based interfacial acoustic wave (IAW) devices.

Strategy	Description
Optimized IDT design	Design the interdigital transducers (IDTs) by optimizing geometry, directivity, and number of fingers to maximize the electromechanical coupling coefficient (*K*^2^) of the propagating acoustic mode.
Minimization of wave reflections	Minimize acoustic wave reflections at the substrate edges and the rear surface of the LiNbO_3_ substrate to reduce energy loss and signal distortion.
Adhesive film control	Ensure proper flatness of the adhesive layer and eliminate air bubbles during the bonding process to maintain uniform acoustic coupling.
Alternative adhesive materials	Consider alternative adhesives such as PMMA (polymethyl methacrylate) or PI (polyimide) to potentially improve mechanical and acoustic properties.
Crystal cut selection	Explore different cuts of the LiNbO_3_ crystal that support interfacial acoustic wave (IAW) propagation with optimal *K*^2^ values.
Alternative IDT materials	Investigate the use of different materials (and their thickness) for the IDTs to enhance electrical conductivity, acoustic performance, and fabrication compatibility.

## Data Availability

The original contributions presented in the study are included in the article, further inquiries can be directed to the corresponding author.

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
