# Peer review of "Insight into the Propagation of Interface Acoustic Waves in Rotated YX-LiNbO3/SU-8/Si Structures"

_micromachines, 2025, doi:10.3390/mi16080861_

Round 1

Reviewer 1 Report

Comments and Suggestions for Authors

The paper numerically and experimentally investigates the interfacial acoustic waves in the structure composed of the rotated YX cuts of lithium niobate, SU-8 layer and silicon. Numerical data are obtained by finite element method and agree fairly well with experimental results.

I think that the article can be published after some revision.

  • Check the notation of the electromechanical coupling coefficient. It is referred to as K2 in places.
  • Last sentence on p.2: “The characteristics of IAWs propagating in multi-layered structures …” It would be logical to add references to relevant publications.
  • It is hard to see difference between images a)-d) in Fig. 5. I recommend that the authors try to find another way to demonstrate their results.

Author Response

Dear Reviewers, thank you for your comments and suggestions.

The manuscript has been revised according to your comments; some text has been added which can be distinguished by the different colour (blue and yellow). Please notice that the text added in the manuscript is not reported in this letter.

Referee 1

1. Check the notation of the electromechanical coupling coefficient. It is referred to as K2 in places.

All the K2 were corrected.

2. Last sentence on p.2: “The characteristics of IAWs propagating in multi-layered structures …” It would be logical to add references to relevant publications.

The authors intended to talk about their present paper.

3. It is hard to see difference between images a)-d) in Fig. 5. I recommend that the authors try to find another way to demonstrate their results.

Figures 5 a-d have been changed and some text has been added.

Thank you

Best regards

Cinzia Caliendo

Reviewer 2 Report

Comments and Suggestions for Authors

In the article “Insight into the Propagation of Interface Acoustic Waves in Rotated YX-LiNbO3/SU-8/Si Structures,” The authors present the numerical and experimental results of interface SAWs (or called IAW) in a multilayered structure. The structure consists of a LN substrate and a Si overcoat with a SU-8 interlayer. Two IAW eigenmodes were discovered, and detailed research was conducted and experimentally verified. This study provides more understanding of IAW and may be interesting in some applications, e.g. acoustic microfluidics. Acceptance of this manuscript should be considered after appropriate revisions. I have the following specific comments, questions and suggestions for authors to improve their manuscript.

  1. Different rotation angles from the Y-axis have been considered. However, the optimal cutting of LN was not applied to the experimental setup. Please comment on experimental considerations and the advantages of the experimental device.
  2. Line 152, the considered loss factor of SU-8 is the same order of LN and Si. I am wondering why SU-8 does not have larger loss factor? Please clarify.
  3. Line 167, in certain range of rotation angle, LSAW and upper IAW disappear. Please explain why these modes do not exist in the range.
  4. In figure 4, the displacement fields of the two IAW modes do not penetrate into the Si, and the fields look discontinuous. Please justify these results.
  5. In the experimental results, figures 10a and 11a, the S21 value of IAW is worse than that of SAW. Why use IAW instead of SAW?
  6. Because of the SU-8 interlayer and Si overcoat, the proposed structure could be promising in microfluidic applications to replace the SAW actuators. Following recent publications related to SAW microfluidics are suggested to be included in the references: https://doi.org/10.1063/5.0134646; https://doi.org/10.3390/mi16060619; https://doi.org/10.3390/s25051577.

Author Response

Dear Reviewers, thank you for your comments and suggestions.

The manuscript has been revised according to your comments; some text has been added which can be distinguished by the different colour (blue and yellow). Please notice that the text added in the manuscript is not reported in this letter.

Referee 2

1. Different rotation angles from the Y-axis have been considered. However, the optimal cutting of LN was not applied to the experimental setup. Please comment on experimental considerations and the advantages of the experimental device.

The best cut (around 80°) is not readily available to the authors.  The experimental devices were fabricated to check the simulation results at least on one rotation angle cut (90° y rot.) which is quite good as well. In future, we plan to design and fabricate optimized device for specific applications using other cut angles.

2. Line 152, the considered loss factor of SU-8 is the same order of LN and Si. I am wondering why SU-8 does not have larger loss factor? Please clarify.

We agree that SU-8 likely exhibits a higher loss factor than the other materials. However, as highlighted in the paper, the loss factor of SU-8 is not reported in the literature, and even its Young’s modulus and Poisson’s ratio are not consistently established. For these reasons, we chose to use the same loss factor for all materials. Nevertheless, as stated in the conclusions, the mechanical properties of SU-8 will be further investigated in order to improve the device design. Some text has been added.

3. Line 167, in certain range of rotation angle, LSAW and upper IAW disappear. Please explain why these modes do not exist in the range.

In a certain range of rotation angle both the LSAW and upper IAW crosses the BAW velocity and, hence, become unphysical solutions, in accordance with the numerical results shown in reference Naumenko. Some text has been added.

4. In figure 4, the displacement fields of the two IAW modes do not penetrate into the Si, and the fields look discontinuous. Please justify these results.

The acoustic field penetrates a little in the Si substrate. We have modified figure 4 in that we have added a picture showing the depth profile of the displacement components to clarify it. Some text has been added.

5. In the experimental results, figures 10a and 11a, the S21 value of IAW is worse than that of SAW. Why use IAW instead of SAW?

Actually, The insertion losses of the upper IAW are lower than those of the corresponding LSAW, while the losses of the lower IAW are larger than those for the SAW. As written in the discussion paragraph, the performance of the device can be improved by using optimized device parameters. In any case, IAW-based devices can be advantageous for microfluidic or package less device applications. Some text has been added.

6.Because of the SU-8 interlayer and Si overcoat, the proposed structure could be promising in microfluidic applications to replace the SAW actuators. Following recent publications related to SAW microfluidics are suggested to be included in the references: https://doi.org/10.1063/5.0134646; https://doi.org/10.3390/mi16060619; https://doi.org/10.3390/s25051577.

We have added to the manuscript the three suggested references which can be useful to the readers who want to get insight the topic.

Thank you

Best regards

Cinzia Caliendo